# *Reviews and Syntheses:* Evaluating the Potential Application of Ecohydrological Models for Northern Peatland Restoration: A Scoping Review

Mariana P. Silva[1], Mark G. Healy[2], Laurence Gill[1]

[1] Department of Civil, Structural, & Environmental Engineering, Trinity College Dublin, D02 PN40, Ireland
[2] Civil Engineering and Ryan Institute, University of Galway, H91 TK33, Ireland

*Correspondence to*: Mariana P. Silva (silvam@tcd.ie)

**Abstract.** Peatland restoration and rehabilitation action has become more widely acknowledged as a necessary response to mitigating climate change risks and improving global carbon storage. Peatland ecosystems require restoration timespans on the order of decades and thus cannot be dependent upon the shorter-term monitoring often carried out in research projects. Hydrological assessments using geospatial tools provide the basis for planning restoration works as well as analysing associated environmental influences. "Restoration" encompasses applications to pre- and post-restoration scenarios for both bogs and fens, across a range of environmental impact fields. The aim of this scoping review is to identify, describe, and categorise current process-based modelling uses in peatlands in order to investigate the applicability and appropriateness of eco- and/or hydrological models for northern peatland restoration. Two literature searches were conducted using the Web of Science entire database in September 2022 and August 2023. Of the final 211 papers included in the review, models and their applications were categorised according to this review's research interests in 7 distinct categories aggregating the papers' research themes and model outputs. Restoration site context was added by identifying 229 unique study site locations from the full database which were catalogued and analysed against raster data for the Köppen-Geiger climate classification scheme. A majority of northern peatland sites were in temperate oceanic zones or humid continental zones experiencing snow. Over one in five models from the full database of papers was unnamed and likely single-use. Key themes emerging from topics covered by papers in the database included: modelling restoration development from a bog growth perspective; the prioritisation of modelling GHG emissions dynamics as a part of policymaking; the importance of spatial connectivity within or alongside process-based models to represent heterogeneous systems; and the increased prevalence of remote sensing and machine learning techniques to predict restoration progress with little physical site intervention. Models are presented according to their application to peatlands or broader ecosystem and organised from most to least complex. This review provides valuable context for the application of ecohydrological models in determining strategies for peatland restoration and evaluating post-intervention development over time.

**1 Introduction**

Peatlands play a vital role in global carbon (C) storage and climate regulation. However, their millennia-long cooling influence is now undermined through human activity, not least the active degradation of extensive areas of peatland and subsequently the effects of climate change (Helbig et al., 2022). In response, northern peatland restoration and rehabilitation activity has increased significantly with large-scale projects in industrial and governmental spheres. Restoration projects and related research often occur in "bursts" with typical spans of 4-5 years based on funding availability (e.g., the European Union LIFE

projects), or collaborations between academic institutions and other organizations. However, peatland ecosystems take decades to millennia to develop and as such, their impact cannot be perfectly extrapolated from the shorter spans within which these projects are carried out (Bacon et al., 2017).

Restoration plans will need to vary in their responses according to the type of peatland degradation that has occurred. Examples of activities having contributed to peatland degradation include drainage for livestock agriculture, especially in northwestern

(NW) Europe, and the creation of oil palm plantations in tropical regions, especially Indonesia. Another example is peat harvesting for fuel or horticulture (where mining can extend down to the mineral soil in places), along with manual cutting along bog margins, which makes site-by-site hydrology extremely variable, especially where not well monitored. Frequently practised restoration strategies arising from peatland research and industrial action include: (1) inundation, achieved by efforts such as drain blocking, bunding, or cessation of pumping, favoured in its simplicity as a direct re-wetting approach; (2) topsoil

removal, achieved by removing of the top layer of degrading soil (often less than 30 cm) and vegetation to mitigate nutrient export/minimise nutrient availability for the formation of new biomass; and (3) slow rewetting, a more controlled and progressive alternative to spontaneous inundation of long-term drained peatlands or costly topsoil removal, which diverge from standard restoration practices (Zak & McInnes, 2022).

When the natural hydrological functions of peatlands are disturbed, the ecosystems react sensitively, with consequences for

whole-catchment hydrology, soil properties, water quality, and biodiversity (Dettman et al., 2014). Environmental impacts of degraded or restored bogs (in their biodiversity, nutrient transport, or C emissions) also depend on site hydrology (Strack et al., 2022). There is an ecological link in the net impact of hydrology on climate change mitigation (as C-storage or C-sinks) which is a primary goal of restoration, alongside efforts to improve regional and nationwide biodiversity: from policy and management perspectives, peatlands are more often considered as nature-based solutions to climate issues based on greenhouse

gas (GHG) emissions, which have concomitant "co-benefits" for biodiversity and water quality (Strack et al., 2022). To reflect this, ecohydrological models include interactions between climate, hydrology, and landscape characteristics, producing outputs describing soil moisture, water level and flow, soil nutrient quantities, sediment transport, or vegetation community patterning/growth (Acharya et al., 2017). From this, the key hydrological functions of peatlands can be linked with climate change-related GHG outputs, or environmental protection/remediation-related water quality outputs, of which the former is a

more prevalent (but no less important) topic in research than the latter according to a dynamic topic modelling study by Yang

et al. (2023), especially considering the perceived urgency of peatland restoration as a response to mitigating climate change (Glenk et al., 2021).

Modelling peatland restoration is useful in its flexibility and in giving the potential to calibrate against different cases of rehabilitation and revegetation. Spatial hydrological assessments using models like MODFLOW (e.g., Brandyk et al., 2016), SIMGRO (e.g., Jaeincke et al., 2010; Povilaitis and Querner, 2008), SAGA (e.g., Ikkala et al., 2022), FLUSH (e.g., Haahti et al., 2016), TOPMODEL (e.g., Goudarzi et al., 2021), DigiBog_Hydro (e.g., Putra et al., 2022), or unnamed numerical algorithms (e.g., Kennedy and Price, 2004; Luscombe et al., 2016) provide the basis for creating a restoration plan, which often prioritises raising the water table as the key engineering goal with outputs confined to groundwater head/flow, surface water level, or catchment runoff/discharge. Yet, despite decades of research, models of this kind are deficient in addressing the entirety of restoring peatlands (i.e., the degradation, mid-restoration, and long-term impact stages) in an efficient, ecohydrological manner.

## 1.1 State-of-the-art models available to date

A variety of hydrological modelling approaches can be used in a northern peatland restoration context, including conceptual models, empirical models, and physical (i.e., process-based/numerical) models (Lana-Renault et al., 2020), and more recently, machine learning (ML) models (Shen et al., 2021). These models are not always ecohydrological in nature but can be manipulated to operate to this end.

Belyea and Baird (2006) present R.S. Clymo's theoretical Bog Growth Model (BGM) as the basis for their conceptual models of peatland development, and call for the consideration of peatlands as Complex Adaptive Systems for future models. The authors found that the primary limitation of the BGM was a lack of accounting for cross-scale coupling of hydrological and ecological processes. They suggested that linking hydrological and ecological processes may provide insight on peatland structure development, while noting that (at the time of publication) feedbacks across temporal and spatial scales could not be properly incorporated (Belyea & Baird, 2006). Further developments to this include a conceptual model of seven hydrological feedbacks specifically linked to WTD: net positive feedbacks related to afforestation/shrubification and specific yield, and net negative feedback related to *Sphagnum* moss surface resistance and productivity, peat deformation and decomposition, and groundwater transmissivity (Waddington et al., 2014). The authors acknowledge that some feedbacks still lack process-level and/or trans-disciplinary, ecohydrological understanding, especially where the ecological focus on *Sphagnum*'s specific impact on hydrology is concerned. Less general hydrological conceptual models have since been developed after extensive site-based monitoring, to inform further predictions of restoration or climate change impacts (e.g., Lhosmot et al., 2023).

Models of peatland dynamics can be specific to climate policy-related outputs rather than providing a full picture of hydrology and ecology. Empirically based statistical GHG estimation models have been developed for C budgeting (van der Snoek et al., 2023; Swails et al., 2022). van Der Snoek et al. (2023) developed a decision support tool (DST) to produce daily, monthly, or annual GHG budgets and a statistical model (SET) to calculate GHG budgets only annually. Swails et al. (2022) use a multiple

regression model to demonstrate responses between soil respiration, soil temperature, and water table depth (WTD) in a restoration context, but only considered these in one site-specific study (in the southeastern United States).

The process-based DigiBog model (Baird et al., 2011; Morris et al., 2012) combines C-accumulation with a hydrology submodule (Young et al., 2017). However, the C-accumulation submodule accounts only for deep peat modification, meaning that there is a discrepancy between rapid near-surface C-accumulation and overall C sinks in a peatland (Young et al., 2019). Recent studies also use ML instead of process-based models, as researchers such as Koch et al. (2023) claim that site-specific knowledge for hydrology, topography, and soil properties makes process-based modelling difficult in large-scale applications.

Active development of new ML models is occurring with increased specificity, as well, such as one developed by Horton et al. (2022) to describe the distribution of fires in tropical peatlands for testing the potential impact of management and restoration scenarios.

Recent efforts have been made to identify available models for peatland dynamics in a review by Mozafari et al. (2023), though the review made little explicit reference to peatland restoration. Additionally, a scientometric review was completed by Apori

et al. (2022) to identify papers that examine the restoration of degraded peatland in the current body of literature, but neglects a focus on modelling.

## 1.2 Knowledge gaps

Restoring site hydrology (i.e., raising WTD) is often considered by engineers and policymakers as the landmark for restoring an ecosystem. Desirable ecohydrological model outputs will depend upon (and likely go beyond) detailed site hydrology

which, in the case of restoration, may create distinct zones of varying hydraulic behaviour contingent upon the scale of degradation or the restoration technique applied. Additionally, process-based models have infrequently been calibrated to describe degraded scenarios or a transition from a degraded to a restored scenario.

Modellers may be required to understand how and when habitats will change, how much new peat may accumulate long-term, and the subsequent C cycle implications of such evolutions, which is difficult to find in any single modelling exercise in

literature to date. There are very few state-of-the-art, process-based, ecohydrological models available to date which have been employed in a peatland restoration application (Mozafari et al., 2023). The potential application of existing models to peatland restoration (considering more than single-state scenarios) has, to date, not been explored systematically. Engineers, planners, researchers, and policymakers would benefit from being more informed about what models currently exist not only to describe peatlands, but to describe peatland restoration, which would be favoured over developing one's own model tailored to a single

restoration scenario or location.

Therefore, the aim of this paper is to identify, describe, and categorise current process-based modelling uses on peatlands in order to investigate the applicability and appropriateness of eco- and/or hydrological models for northern peatland restoration. This will unite ecohydrological and restoration modelling interests by interpreting the relevance of different models' output(s) to peatland restoration projects, with the expected goal to predict the success of restoration measures undertaken in northern

peatlands. Hence, the exploratory nature of this review is scoping rather than systematic.

## 2. Materials and Methods

### 2.1 Literature searches: September 2022 and August 2023

A literature search was conducted using the Web of Science entire database in September 2022 and updated for the 2022-2023 period with a second search in August 2023. The following search string was employed in an advanced search: TS= (peat* AND ((model* AND hydrolog*) OR (model* AND GHG))). Note that there remains a significant knowledge gap in water quality modelling (Yang et al., 2023), such that GHG modelling was emphasised in this review, and no equivalent search term was applied for water quality. It was anticipated that the term "hydrolog*" would be sufficient to encompass water quality and nutrient transport modelling if these papers existed in the literature.

The combined search yielded 1116 results. These references were downloaded in *.ris* format and collated in the free, open-source reference management software, Zotero (Corporation for Digital Scholarship, George Mason University). The papers were screened three times in alphabetical order, with each iteration eliminating inappropriate titles and reducing the batch size. While some papers in the search overlapped with the previous search, regular comparisons with the existing database were heeded to avoid double-counting. This addition to the full dataset is shown in Table S3.

### 2.2 Determination of appropriate models

Models were considered for their potential application in an ecohydrological manner, despite differences in the actual processes incorporated in individual model codes, and whether or not they are strictly called ecohydrological (e.g., a purely hydrological model being shown to operate well in a peatland context). As it may be valuable to investigate long-term bog growth for systems so fundamentally changed by mining or degradation that returning to a "former" habitat is not likely, modelling organic matter decomposition and emission over the span of decades or centuries is pertinent for consideration beyond current hydrological and C dynamics. It is unlikely that any single model will be able to encapsulate all these interests for simulating a restored bog or other northern peatland region. Therefore, it was valuable to document which modelling options exist on local to global scales, as well as in different dimensions of space, time, or conceptual structure, and how these models operate in case some can be combined or used in tandem.

A first pass considered all paper titles, and occasionally portions of paper abstracts for clarity. A second pass was made where the methods sections of each article were read in full. This was done to identify which model interfaces are used and discard those papers that do not name any models or whose models' functions do not match the scope of this review. Criteria for the rejection of a number of articles fell into two themes: too much specificity, or a lack of connection with hydrology (Table 1). This review focusses on "northern" peatlands, existing above 50-40°N (with some emphasis placed on NW Europe, given this is a region of interest for the authors), as they are distinct enough in climate and land use contexts from "tropical" peatlands such that modelling in these categories is generally known to diverge in literature (Tarnocai and Stolbovoy, 2006). Some tropical peatland hydrology studies were retained that employ restoration-related modelling which may be useful to identify, despite differences in climate and ecosystem characteristics.

A third and final pass was made during which pertinent information from the full articles was gathered to fill a summary table for each paper. The table, created for each model category, included the following subheadings: (1) Sole focus on hydrology (Pure Hydrology); (2) Focus on greenhouse gases and connected biochemistry (GHG Dynamics), with the ideal to prioritise process-based models which went beyond calculating net ecosystem exchange or global warming potential as sole outputs; (3) Description of long-term peat accumulation projections or reconstructions (Peat Accumulation); (4) Regional or national scaled-up models involving northern peatlands looking beyond a site-specific scale (Global Models); (5) Multiple models from previous categories used in tandem or in sequence (Model Combinations); and (6) Models integrating or coupling processes included in previous categories (Coupled Models). Pertinent information from the second literature search was added to a separate table with identical headers to the original database except for an additional column "Category" linking the papers to the previously generated model categories (five papers appeared relevant but did not neatly fit into any of the existing model categories; their sole focus covered restoration-related remote sensing topics and were categorised as (7) "Remote sensing" to reflect this). Studies performed by the same first author(s) on the same or similar site(s) were considered together as a "suite" during analysis but remained separate for subsequent tables and figures.

A final total of 211 relevant papers was retained in the database. The database was manually reviewed for key models of interest and their relative frequency within the full set (in which 224 models are identified for the 211 papers in the database). Considerations for making this selection were as follows:

1. If coupled model packages (like CoupModel, housed in its own interface, or DigiBog/MPeat, where code is freely available in common languages like Fortran and MATLAB (though varying in cost barrier)) already exist that appear flexible enough to apply to a northern peatland context, these should be prioritised, especially considering the FAIR (Findable, Accessible, Interoperable, and Reusable) accessibility guidelines which can improve research communication and future application of models beyond academia (Mozafari et al., 2023).

2. As demonstrated in Table 2, just over one fifth of the total database consisted of unnamed, often numerical models, hand-made for specific research needs and rarely used more than once or twice. They are therefore less likely to contribute significantly to the sharing of modelling methods because their accessibility is limited to the creator. Additionally, the consolidation of approaches improves confidence in modelling, making the use of unnamed models less beneficial, along with less commonly used models serving similar purposes to more commonly used ones.

3. Focussing on the ecological side of ecohydrological modelling became pertinent because of the wealth of background already present for pure hydrological models and this review's emphasis upon moving beyond hydrology as the priority for restoration targets.

The full dataset, including each paper's DOI, first author, year published, model(s) used, and a brief description of the research scope, is summarised completely in Tables S1 and S3. Within the 211 papers recorded, 229 unique study site locations were identified, catalogued, and analysed against raster data for the Köppen-Geiger climate classification scheme (Tables S2 and S4): developed by Wladimir Köppen and Rudolf Geiger and published by Köppen in 1936, the system has since become an established multidisciplinary standard for describing the climate of a region (Rubel et al., 2017). This was done in order to

evaluate the potential prevalence/preference for the use of different models depending on the climate conditions of sites on which they were tested. Additionally, data types and durations used in some papers were documented as the review progressed, to provide a fuller picture of the potential utilisation of key models.

Post-hoc analysis of shortlisted models occurred to explore what information will aid potential modellers in determining either the suitability of a single model for their purposes, or the possible compatibility of combining existing codes or running models in tandem. This centred around general model specifications and information about data inputs as discussed above. Observations and conclusions made based on data solely from the second literature search will be designated as such.

## 3. Results

The following observations were made for papers collated within the 7 model categories:

**"Pure Hydrology":** Seventy-seven total papers were catalogued for this category, making it the largest (approximately 36.5% of the full database). MODFLOW featured here prominently, with 15 papers using the model, around 19.5% of all models in this category. Only 6 papers across the entire category (7.8%) were published in 2022 or 2023, 3 of which came from the second literature search. Two of the four total water quality-related papers featured in this category (Sutton and Price, 2022;

Nieminen et al., 2018).

**GHG Dynamics:** Forty-one total papers were found for this category, approximately 19% of the full database. Additionally, it is possible that some models in this category are in fact "integrated" or "coupled" (i.e., they include hydrological or other processes, as well as the necessary biochemistry) but the only outputs are GHG-flux related. Ten papers across the category (25%) were published in 2022 or 2023, 9 of which came from the second literature search.

**Peat Accumulation:** Ten total papers were found for this category, making it the smallest (approximately 5% of the full database). The majority of models use historical climate reconstructions as the basis of the model. The papers in this section also had an average publication year of 2009, making this category "older" than the others, all with average ages between 2014 and 2016. No new peat accumulation topics were identified in the second literature search except for one paper classified as a Model Combination which incorporates peat accumulation.

**Global Models:** Sixteen total papers were found for this category, approximately 7.5% of the full database. While it was not expected for many of these papers to be relevant to the current study, because individual countries may appear as only 5–10 cells within a large grid and individual peatland sites may be indistinguishable, it may still be important to account for the models used here, especially those which represent "northern peatland" condition. Only one paper in this category was published since 2021; all others were published in 2020 or earlier.

**Model Combinations:** Nineteen total papers were found for this category, approximately 9% of the full database. "Model combinations" either present outputs not encapsulated by a single model or include the processes from two or more models to yield a more robust output. There may be more combinations among the other categories which are not presented outright, but these papers were considered to be unique because of the combinations they employ. Eight papers across the category (42%)

were published in 2022 or 2023, 6 of which came from the second literature search. The other two of four water quality-related
papers featured in this category (Xu et al., 2020; Bernard-Jannin et al., 2018).

**Coupled/Integrated Models:** The second-largest category (approx. 20.5% of the full database) consists of coupled models housed within the same interface or code, with 43 total items. Some "suites" of papers from the same authors occur here where model development can be tracked, or multiple outputs are analysed for the same sites and published separately over a number of years.

**Remote sensing:** Finally, an emerging trend regarding models that include a remote sensing aspect was found in the past year (2022-2023): almost one in three (9/29) papers from the second literature search included remote sensing data or was based on a remote-sensing classification model. Five of these papers (Ball et al., 2023; Dabrowska-Zielinska et al., 2022; Dadap et al., 2022; Jussila et al., 2023; and Puertas Orozco et al., 2023) presented their approach and results as distinctly remote-sensing oriented and were placed in their own "Remote sensing" category for post-hoc discussion. These take up the remaining 2.5%
of the total database.

Thirteen of the 31 Köppen-Geiger classifications were represented by the peatlands modelled in this review (Figure 1). Coupled models were applied in the largest quantities for the three most abundant classifications (Dfc, Dfb, and Cfb, in that order). It was only in the Cfb region where every identified model category is applied; the Dfb region did not host any remote sensing research and the Dfc region did not host any peat accumulation research. Note that Dfc (subarctic climate), Cfb (warm-summer
temperate oceanic), and Cfc (cool-summer temperate oceanic) classifications describe most of NW Europe, and Dfb (warm-summer humid continental climate) primarily describes Canadian sites which are continental rather than coastal, experiencing more snow. A GIS map layout plotting the coordinate locations of individual study sites is included in Figure 2. For the three least commonly studied climate region classifications (Cwb, Csa, and Dfa), models used fell into the Coupled Models and GHG Dynamics applications, especially with a focus on methane emissions (Wania et al., 2010; Walter et al., 1996).

Seventeen papers did not use physical study sites, many of which were global or regional studies scaling up data from a wealth of site-specific research. Three papers (Bechtold et al., 2020; Qiu et al., 2018; and Qiu et al., 2019) catalogued too many sites to document meaningfully (these were also housed within the "Global Models" category).

The following process-based models had the highest frequency of use within and across model categories (Table 3): MODFLOW (Pure Hydrology); DigiBog, including DigiBog_Hydro (Peat Accumulation, Pure Hydrology, and Coupled
Models); CoupModel (Coupled Models); *ecosys* (Coupled Models and GHG Dynamics); McGill Wetland Model (MWM), including CLASS3W-MWM (GHG Dynamics); PEAT_CLSM (Global Models); and LPJ, including LPJ-GUESS and LPJ-WHyMe (GHG Dynamics). No model was used more than twice in either the Model Combinations or the Peat Accumulation categories. Note that MODFLOW, a household name in hydrological modelling for decades, was not used in model combinations or for any applications beyond "Pure Hydrology" in this review; other Pure Hydrology models which were used
in Model Combinations include PERSiST and TOPMODEL.

The highest dimension attempted by simulations outside of "Pure Hydrology" occurs only once in this review for 3-D (*ecosys:* Grant et al., 2017) and once for pseudo-3D (DigiBog_Hydro: Putra et al., 2022). Examples of 3-D modelling more common

with Pure Hydrology include a MODFLOW model from Sutton and Price (2022)—which also includes hydrochemical transport—and a GEOTop model from Zi et al. (2016).

## Discussion

The most-featured models arising in this review's database are discussed here along with models which may have appeared less often, to present an overview of current model options and identify trends in order of process complexity. Note that "process complexity" here is interpreted as: the degree to which a process is difficult to observe, understand, or explain (McDonnell et al., 2007). This is different from simply capturing the *number* of processes (especially when trying to attribute and discretise an equation for each) in different models, but these properties are often connected.

### 4.1. Peatland specific models: most to least complex

Within past decades, numerical models have been formulated and refined to represent peatland-specific complexities, especially considering changes occurring in past designations of "acrotelm/catotelm" layers (Clymo, 1978), incorporating characteristics interacting with hydrological boundary conditions (such as bog shape) and addressing feedbacks specific to peat decomposition (Morris et al., 2011).

**McGill Wetland Model:** Wu and Roulet (2014) used CLASS3-MWM, a coupled version of the McGill Wetland Model (MWM), to simulate GHG dynamics across multiple climate scenario projections. The 1-D model represents C fluxes as net ecosystem production (NEP) for representative single-year batches across a century, and was able to support fen-bog transitions from circa 2000 to 2100. With imposed climate change scenarios, the model has predicted resiliency in bog ecosystems and that fen-bog transitions could make peatland ecosystems more resilient in a changing climate. The threshold between a MWM-modelled C sink/source was shown in the balance between gross primary production and decomposition, concluding that the soil C response of bogs to climate change is determined chiefly by vegetation production and decomposition in the acrotelm (Wu and Roulet, 2014).

In the vulnerable years at the start of bog restoration, models may need to represent vegetation growth and associated ecohydrological changes with greater detail than what is demonstrated with older versions of MWM, but this has been addressed in two newer versions of the model MWMmic and MWMmic_NP (Shao et al., 2022a, 2022b). With the addition of a multi-layer cohort structure, peat decomposability now decreases with peat depth (Shao et al., 2022a), allowing for the modelling of degraded, formerly deep-peat layers to decompose and release nutrients distinctively from ideal, less consolidated "natural" peat cases. Further, vegetation growth and competition (moss and shrubs) has been linked with hydrology and nutrient availability in newer versions (Shao et al., 2022b), improving the ecohydrological character of the model (Figure 3). MWM is not publicly available, and it is not stipulated how to request access.

**Peatland-VU and PVN:** Peatland-VU and its contemporary, Peatland-VU-NUCOM (PVN), have been developed to model peatland GHG emissions and dynamic vegetation (Lippmann et al., 2023). It presents itself similarly to CoupModel discussed

below (both are 1-D), except abridged to a peatland context and with a focus on emissions outputs. Parameters do not focus on complex hydrology (beyond requiring WTD input data), and vegetation is organised as plant functional types (PFTs) with variable relative biomass quantities depending on competition for light relative to parameter-defined growth requirements. The model is free and available as Fortran/C++ source code on Zenodo (https://www.bitbucket.org/tlippmann/pvn_public, last accessed: 27 February 2024).

**DNDC and Wetland-DNDC:** Wetland-DNDC (DeNitrification-DeComposition) appeared in the coupled models section twice, and the core DNDC model appeared in the GHG dynamics section twice. Wetland-DNDC has been superseded by Forest-DNDC (Gilhespy et al., 2014) more often used in boreal forested peatlands (Kim et al., 2016), though Wetland-DNDC is still being used occasionally (e.g., Mikhalchuk et al., 2022). The models focus on C and nitrogen (N) cycling with an original focus on agriculture with uses in natural contexts (Webster et al., 2013). The primary focus is on biochemical processes, requiring common soil chemical/hydrological and climate inputs, as well as vegetation variables in the form of "crops". Wetland-DNDC and Forest-DNDC considers wetland restoration as a major management practice and have been parametrized to represent this; however, the models are not physically based and requires numerous parameters for microbially-mediated chemical transformations.

**LPJ-WHyMe:** Originating from the Lund-Potsdam-Jena model (LPJ), this version simulates wetland hydrology and methane dynamics for global, regional, or site-scale vegetation modelling (Wania et al., 2013). It was developed specifically for understanding permafrost dynamics on northern peatlands and then expanded to consider methane emissions. The model uses PFTs as the conceptual basis for vegetation growth, with a simplified focus on wetland vegetation and a separate category just for *Sphagnum* moss. The model code (Fortran77) is available by request through the Oak Ridge National Laboratory. The model takes in meteorological information, soil type and land/ocean mask maps, as well as text files for climate, vegetation, methane and soil relevant variables, and parameter values. The model appeared less often than LPJ-GUESS (discussed below), and while they have a shared conceptual basis, they have since diverged. Recent works not captured in this review have used LPJ-WHyMe in non-wetland or partially-wetland scenarios (Huang et al., 2024; Sun et al., 2020; Sun and Mu, 2022), possibly due to its permafrost dynamics capabilities.

**ELM-SPRUCE:** the ELM-SPRUCE model was developed based on the Energy Exascale Earth System Model (E3SM) and Community Land Model (CLM) for use in Oak Ridge National Laboratory's SPRUCE experiment on peatlands in Minnesota, USA (https://mnspruce.ornl.gov/, last accessed 6 March 2024). It is specific to the boreal peatland ecosystem on which it was developed, with a particular focus on improving spatial methane dynamics, to be integrated back into the E3SM global model as an ultimate goal (Yuan et al., 2021).

**PEAT-CLSM:** Bechtold et al. (2020) developed a northern peatland-specific land surface hydrology model to NASA's Catchment Land Surface Model (CLSM), using a TOPMODEL approach (discussed below), where microtopography as hummocks and hollows are emphasized along with peatland-specific vegetation (especially in surface water ponds) and peat soil hydraulic properties are incorporated as parameters. While the peatland-specific improvements are simple to conceptualise, computational units on the global scale (as catchments, which can be discretised into grid cells) introduces complexity. The

model is available as sections of NASA's CLSM *catchment.F90* code that were modified to become PEAT-CLSM modules (https://osf.io/e58ym/, last accessed 6 March 2024).

**Modelling hummocks and hollows:** several papers have produced numerical models specifically to capture dynamics in hummock-hollow patterning (including ridge-slough dynamics in the Florida Everglades). These were largely unnamed (Couwenberg, 2005; Eppinga et al., 2009; Heffernan et al., 2013; Kaplan et al., 2012), with two named models arising. One is HOHUM (Nungesser, 2003), though it has not been expanded upon or applied in literature since its inception; the other is Hummock-Hollow (HH) (Cresto Aleina et al., 2015; 2016), which integrates the relationship between microtopography and

emissions fluxes – for which the theory has been cited in papers using ELM-SPRUCE (Ricciuto et al., 2021) and ORCHIDEE (Yao et al., 2022) models, but not directly applied in new scenarios.

**MPeat and MPeat2D:** MPeat is similar to HPM and DigiBog but includes peat mechanical processes within a 1-D peat column. It introduces more complex relationships for bulk density, active porosity and hydraulic conductivity, as well as including Young's modulus as an additional variable (Mahdiyasa et al., 2021). A newer version, MPeat2D, has expanded on

the spatial capabilities of the model, as well as adding an ecological submodule, but with limited application yet due to its recent development (Mahdiyasa et al., 2023). It is free and available as MATLAB code (https://zenodo.org/records/10050891, last accessed 6 March 2024).

**Holocene Peat Model (HPM):** most prominently used for reconstructions of peatlands in past climates, HPM was applied to learn more about the nature of peat accumulation against C-dated peat cores (Quillet et al., 2013; Liu et al., 2018). It is 1-D

(where it is meant to model the centre of an ombrotrophic peatland) and uses 12 PFTs from which productivity and decomposition are balanced to develop rates of peat accumulation, along with some peat physical properties such as variable bulk density. The model was used in a comparison with MPeat discussed prior (Mahdiyasa et al., 2021), and has not been made publicly available except in a modified Arctic permafrost version by Treat et al. (2021) (https://zenodo.org/records/4647666, last accessed 6 March 2024).

**Digibog and DigiBog_Hydro:** the peat accumulation model DigiBog is particularly concise when considering ecosystem functioning, choosing to represent the plant growth process through the relative rates of litterfall and decay to the atmosphere (Morris et al., 2012). The 2-D application of DigiBog for a blanket peatland, demonstrated by Young et al. (2017), has been used to investigate the impacts of peatland drains and drain blocking (Figure 4). In some DigiBog simulations, water table can be fixed or forced before evaluating bog response, allowing for WTD predictions to vary based on human interventions (Young

et al., 2017). It should be noted that a constant bulk density is used in DigiBog.

DigiBog_Hydro has connectivity with GIS for determining peatland-specific hydrology in a 2-D plane, with existing applications to a restoration context (Putra et al., 2022). The required inputs for DigiBog_Hydro are entered into a GUI, so no coding knowledge is required unless further modifications are attempted. Additionally, the user manual is well put together and freely available. DigiBog_Hydro is a submodel of DigiBog and thus neglects some of the peat accumulation elements of

the full DigiBog model (and similarly assumes a constant bulk density), though it can provide a starting point to obtain a hydrological "snapshot" for a site area before (or concurrently with) connecting to peat accumulation or ecosystem dynamics

over time. Both models are free, where DigiBog is publicly available and DigiBog_Hydro is available by request (https://water.leeds.ac.uk/our-missions/mission-1/digibog/resources/, last accessed 6 March).

### 4.2. General models: most to least complex

*ecosys:* the output capabilities of the fully coupled mathematical ecohydrological model *ecosys* overlap with many other models of interest. Its extreme process richness requires explicit understanding of the theory governing plant substrate and microbial populations which acquire, transform, and exchange resources (as energy, water, C, N, and phosphorus (P)), and computational power for simulations in 1-D, 2-D, or 3-D, with multiple canopy ad soil layers, and with sub-hourly timesteps (Figure 5; Grant, 2013). The model is free with documentation available by request (https://github.com/jinyun1tang/ECOSYS,

last accessed 6 March). *ecosys*'s complexity does not make the model a "better" choice inherently; in fact, the model over- and under-estimated seasonal GHG fluxes similarly to less complex models across multiple land cover types, perhaps calling into question the necessity of such detail if outputs are not significantly more accurate or precise (Sulman et al., 2012). For the purposes of evaluating ecohydrology, however, *ecosys* matches performance from the ecological process side while also simulating WTD, unlike other models reported by Sulman et al. (2012) which do not have hydrological outputs.

Dimitrov et al. produced *ecosys* models for a bog (2010) and later a peatland transition zone (2014) focussing solely on hydrological outputs. Models did not display very high accuracy ($R^2$ between 0.40 and 0.56) for monitoring WTD and water content in the intact bog, but were even less accurate without including a macropore flow process ($R^2$ between 0.27 and 0.41). Model applications have since expanded to fully coupled ecohydrology in both peatland and non-peatland contexts. Mezbahuddin et al. (2017) used *ecosys* for coupled ecohydrological modelling in a Canadian boreal fen, controlling lateral

flow between a chosen number of cells, while including microbial activity and carbon dioxide ($CO_2$) fluxes (which can be reduced to GPP or NEP), as well as simulating WTDs within the plane, with decent accuracy ($R^2$ between 0.68 and 0.84 for $CO_2$ fluxes).

**MODFLOW, MODFLOW SURFACT, and MODPATH:** a 3-D finite-difference groundwater flow model with added vadose-zone capabilities, MODFLOW is well known for being developed by the USGS and frequently used and improved

upon; indeed, the most recent release MODFLOW 6 does include an unsaturated zone flow package but still focuses on groundwater transport (https://water.usgs.gov/water-resources/software/MODFLOW-6/, last accessed 6 March 2024). Input and output parameters are too numerous to list, and the model requires knowledge of finite-difference cell-based discretisation concepts; within the field of pure hydrological modelling, there may be a prerequisite for knowledge in computational modelling, but MODFLOW's demands may not best combine with general ecosystem process modelling. MODPATH

demonstrates water quality connectivity, though none of the papers using MODPATH on peatlands in this review simulate water quality (Reeve et al., 2000, 2001a, and 2001b; Reeve and Gracz, 2008; Reeve et al., 2009). Older versions of MODFLOW such as MODFLOW SURFACT may prove simpler to use for peatlands where the vadose zone is more critical than groundwater flow (Sutton and Price, 2022).

**ORCHIDEE:** the global land surface model ORCHIDEE consists of a trunk version for global applications of water-energy-C budgets as well as branches for more detailed simulations such as high latitude peatland C dynamics (Largeron et al., 2018; Qiu et al., 2019; Kwon et al., 2022). The model is free and available as Fortran code (https://orchidee.ipsl.fr/you-orchidee/, last accessed 6 March 2024), though with extensive computing power and data inputs required which are too numerous to name here.

**CoupModel:** this 1-D, fully integrated process-based model can simultaneously simulate WTD and $CO_2$ fluxes based on water, soil organic and inorganic, and canopy/groundcover vegetation energy processes (e.g., Kasimir et al., 2018; He et al., 2023a, 2023b). Like *ecosys,* CoupModel requires a rigorous understanding of manifold chemical and energy transformation processes, but CoupModel is housed within a browser User Interface (UI) rather than requiring coding knowledge, where visualisation of the soil column and water/energy fluxes is simple but available to the user. The model provides numerous "switch" specifications for selecting if certain processes should be considered and in what manner (e.g., choosing to include snowmelt processes or soil freezing and thawing, or neither) to determine the ultimate number of parameters necessary in a given simulation. While some default parameters exist, and soil/vegetation categories based on a past simulation database, the total number of parameters is still extensive.

**Soil & Water Assessment Tool (SWAT):** As an ecosystem-based model, SWAT has been adapted for uses ranging from farmland nutrient tracing to cold wetland watershed organic cycling (Kalcic, Chaubey, and Frankenberger, 2015). It is 2-D and formed on the basis of Hydrologic Response Units (HRUs), which are user-defined areas of similar land use, soil type, and topographical characteristics. The watershed conceptualisation is configured via spatial objects, differentiating HRUs and collecting them into a "landscape unit", with a "routing unit" connecting delineated aquifer(s), channel(s), and reservoir(s) in the landscape. The model requires upwards of thirty input files; a pre-existing database provides groups for plants and Generic Land Covers and a number of associated parameters, and little to no pre- or post-processing is required outside of the interface, similar to CoupModel. SWAT executables as command-line code, as well as GIS-integrated interfaces ArcSWAT and QSWAT, are publicly available on the SWAT website (https://swat.tamu.edu/, last accessed 6 March 2024).

While it did not feature as often in frequency in a peatland context, the SWAT model modified by Melaku et al. (2022) adds a wetland-specific emissions subroutine which estimates groundwater table, $CO_2$ emissions, and net ecosystem exchange spatially and temporally. The number of currently incorporated HRUs limits the resolution available for classifying different kinds of wetlands, which may pose a challenge for demonstrating heterogeneity in site-scale modelling (Melaku et al., 2022).

**LPJ-GUESS:** LPJ-GUESS is a dynamic global vegetation model designed for a global to regional scale incorporating plant physiology of "individuals" within a stand (as PFTs), as well as population dynamics and soil biogeochemistry (Figure 6). It was originally developed for bioclimatic zones where entire countries may be represented by a single set of PFTs (Smith et al., 2001). The model code (C++) is publicly available (https://web.nateko.lu.se/lpj-guess/download.html, last accessed 7 March 2024). Tang et al. (2018) added DOC and "water routing" modules to the model; they simulated dissolved organic carbon (DOC) transport through a watershed (excluding gaseous C fluxes) using LPJ-GUESS, demonstrating higher DOC

quantities in peatlands (bog and fen) than in mineral soils. However, there is no indication of the consideration of peat growth or microtopography, though litter and soil organic matter turnover is an included process.

**SIMGRO:** the mechanistic-distributed SIMGRO model is a combination of MetaSWAP land surface/unsaturated zone and MODFLOW phreatic zone models, housed in a GIS interface. The model is not publicly available, and it is not clearly stipulated if downloads can be requested (https://www.wur.nl/en/research-results/research-institutes/environmental-research/facilities-tools/software-models-and-databases/simgro.htm, last accessed 6 March 2024). SIMGRO is able to generate a 3-D spatial representation of WTD rise pre- and post-restoration across an entire domed peatland area (Figure 7), though with no connection to carbon cycling or peat accumulation processes (Jaenicke et al., 2010). The simulated hydrology was related to C storage by providing an index (developed by Couwenberg et al. in 2009) for C stored in a peatland of a given area per centimetre of groundwater rise. While this model does not feature prominently in this review or within recent northern peatland applications, SIMGRO remains one of the few models in this review which carries out a multi-dimensional representation of peatland restoration specifically.

**GEOTop:** GEOTop, a contemporary of TOPMODEL, is a watershed-distributed hydrological model which couples water and energy balances. It is free and available as Fortran code through a web repository (https://github.com/geotopmodel/geotop, last accessed 6 March 2024). Inputs include elevation as a Digital Terrain Model, soil-type mapping, land-use mapping, and hourly meteorological time series data. The representation of vegetation growth as energy flux (i.e., net radiation, sensible, latent, and ground heat fluxes) calls for inputs of landcover maps with fixed vegetation zones, and solving the energy equation in 1-D despite solving the water balance in 3-D (Zi et al., 2016).

**TOPMODEL:** TOPMODEL is a topographically-based model describing the dynamics of surface and subsurface saturated areas using simplified storage-discharge relationships in steady state. A grid cell size of <=50 m is recommended, so it can be flexible for smaller catchments – and yield a good idea of water transport and its connectivity with water quality. Hydrological and topographic inputs are all that are required, though outputs are only in the form of a hydrograph/sensitivity analysis and so remains distinct from any ecological processes. It is free and available online as Fortran code (https://cran.r-project.org/web/packages/topmodel/index.html, last accessed 26 February 2024). It was used in 2006 in tandem with InTEC 3.0 as a simple hydrological input to model GHGs on a multi-year, national scale on peatlands (Ju et al., 2006), and has been used more recently in more pure hydrological scenarios (Beven et al., 2021).

**Modelling products (Visual MODFLOW© and HydroGeoSphere):** Visual MODFLOW has simplified much of the set-up work and visualisation by framing MODFLOW (including the newest version) in a GUI for groundwater flow and solute transport only (used in Brust et al., 2017; see specifications at: https://www.waterloohydrogeologic.com/products/visual-modflow-flex/; last accessed 8 March 2024). However, this is a modelling *product* and exists behind a paywall, diverging from FAIR modelling ideals. HydroGeoSphere (HGS™) is a fully-integrated 3-D finite element software carrying out similar modelling ends to MODFLOW except with a node/element mesh rather than with cells (https://www.aquanty.com/hgsfeatures). Jaros et al. (2019) use HGS™ alongside a global sensitivity analysis techniques reduce the number of parameters required to model groundwater-surface water interactions in aapa mires. While the physically-

based model produces spatially-intuitive results, it may require a separate software for visualisation on top of being a product behind a paywall, again diverging from FAIR modelling ideals.

**HYDRUS:** being housed in a GUI (Windows application), but hidden behind a paywall, the HYDRUS program numerically solves the Richards equation for variably saturated flow, most commonly used in pure hydrology applications (e.g., Hokanson et al., 2021) with abundant applications in non-peatland-specific scenarios. Available in 1-D, 2-D and 3-D, the software is often used in solute transport applications (Šimůnek et al., 2024). Vegetation is modelled as "crops" in the program insofar as to inform water and solute uptake through the root zone rather than considering changes in vegetation dynamics (Šimůnek et al., 2024). The application of HYDRUS to peatlands can be done in scenarios of controlled vegetation and topography, to investigate other properties of the peatland's functioning.

**PERSiST and INCA-C:** this model combination was used in two separate papers captured in this review (Xu et al., 2020; de Wit et al., 2016) with precedents for this combination made by Futter et al. in 2007, 2009, and 2014. The combination captures carbon cycling with INCA-C and watershed hydrology with PERSiST, both models requiring very few measured data inputs – a total of three daily time-series datasets, with additional calibration and validation site measurements required for soil carbon or DOC (Xu et al., 2020). INCA-C conceptualises the carbon cycle by simulated transformations between C pools and fluxes between water pools, organised into organic soils, mineral soils, and open water categories; PERSiST is a derivative rainfall-runoff model with specific focus on solute transport (Futter et al., 2014). $R^2$ values ranged in evaluation periods from 0.44 to 0.78 for simulated discharge and from 0.29 to 0.69 for DOC (Xu et al., 2020), with an even better performance by de Wit et al. (2016) of 0.85. Considering that key hydrological outputs had decent accuracy with significantly less computation required, taking a less process-complex approach may be beneficial in cases where desired outputs are few. However, this approach hinders the ability to investigate the results' implications on other processes in the natural system not parametrized in the model (e.g., the DOC quantities simulated here cannot inform as much about vegetation dynamics or decomposition).

### 4.3. Purely data-driven models

This review did not focus on statistical or empirical models in this review in favour comparing the complexity of process-based models and the potential for representing peatland restoration from a bottom-up, theory-based approach. However, it is notable that machine learning models applied to peatland restoration, while not having a foundation in key environmental processes on the ground, featured with increasing frequency. Models with direct links to spatial imagery software can more easily incorporate already-existing satellite data or mapping data to hopefully reduce process complexity. The five papers which stood apart from the pre-existing analysis' model categories included outputs centred around predictions of soil moisture, vegetation communities/extents, and resulting gross primary productivity estimates; while these outputs have hydrological and ecological themes, they do not directly incorporate hydrological or ecological processes, and rather favour using the existing body of research along with ML predictions to form conclusions about the nature of peatland ecosystems with little to no physical intervention. Four other recent (2022-2023) papers gathered in the Model Combinations and GHG

dynamics categories also featured satellite data with ML classification, though these were treated as data inputs to other models rather than being the end goal of the research.

The challenge posed in bringing together similar types of ML code for this type of review is a lack of official title for these programs apart from well-known algorithms like Random Forest (e.g., in Kou et al., 2022; Rissanen et al., 2023; Ross et al., 2023). Until now, there is not much consolidation of data or codes used for remote sensing projects. As the field of ecohydrological modelling progresses, the connection with spatial imagery may become more prominent and favoured not only as a way to incorporate more data into models when site-level instrumentation proves too difficult to orchestrate, but as

a standalone technique for predicting hydrological and ecological changes and especially for enacting regional or nationwide policy decisions.

## 4.4. Synthesis

The most complex models presented are generally less specific to peatlands, and are fully coupled (e.g., *ecosys)*, which require a large input of data and knowledge to achieve the flexibility to model as many physical or chemical outputs in as many

landscapes as possible. However, general ecosystem models (most often applied for GHG Dynamics or Global Models on peatlands) may be just as complex as some pure hydrological models, where each model may generate entirely separate outputs with few common inputs. Challenges with *ecosys* and many general ecosystem process-based models remain that simulating changes in peat near-surface characteristics (i.e., key features such as bulk density, macroporosity, and vegetation) over time due to restoration is not integrated and may need to be forced, and these programs do not account for peat volume change in

the long term. A key trend in applicability of many of the GHG Dynamics, peatland-specific models (e.g., MWM, PVN, and some applications of CoupModel) is the representation of whole peatland ecosystems with 1-D points where studies may have compared bog and fen ecosystems or climate scenarios, etc., broadly.

There is a considerable lack of recent focus upon peat accumulation modelling as demonstrated by this review, especially by the absence of this category in recent (2022-2023) papers; and, where peat accumulation modelling was studied, these

510 simulations existed mostly in Cfb and Dfb regions, with few other sites. A limited representation of climate regions here does not necessarily imply a lacking global picture of peat accumulation research; however, palaeoecological peat accumulation rate research contributing to scientific understanding about worldwide peat accumulation is not considered in this review. It is only observed that the simulation of peat accumulation (especially for future projections) worldwide is scarce. In studies using peat accumulation models (e.g., HPM, MPeat, DigiBog), changes in soil characteristics (especially where bare peat

rehabilitation is concerned) over time have not been investigated. Though this transition in hydraulic behaviour has been studied in laboratory-scale models (e.g., Gauthier, McCarter, and Price, 2018), and in the field (e.g., Lehan et al., 2022), it has yet to be demonstrated by the process-based models in this review such that the representation of bare peat rehabilitation may still pose a challenge for modelling. They also does not include any of the biogeochemical changes associated with fen-bog transition, which occur along geologic time scales when the water table becomes consistently high to cease meaningful

interactions with groundwater (Wu & Roulet, 2014), or more quickly due to sudden drops in WTD (e.g., volcanic impacts

(Loisel and Bunsen, 2020) and intensive peatland rehabilitation works (Malloy and Price, 2014)) in what would appear in many peat accumulation models as a mere fraction of the spin-up and simulation time scale.

In modelling degraded and rehabilitating landscapes, for which there is little precedent, further field measurements and model modifications will be needed to model the dynamics of new vegetation communities arising in these conditions. He et al. (2023a) recently modelled GHG emissions coupled with hydrological and chemical impacts of an actively extracted peatland using CoupModel, establishing a precedent for this interface. A large number of (especially coupled ecohydrological) models used PFTs to categorise vegetation parameters more simply, either being fixed to the locations initialised or dynamic and competing for resources throughout the simulations. Additionally, setting up land-use boundaries for HRUs may be a useful way to delineate restoration techniques within a bog site or to examine interactions with nearby agricultural lands, but significant assumptions would need to be made in creating such classifications.

Given that many restoration actions control the water table primarily, such as cell bunding managed by weirs or pipes, it is a valuable application of many of the models simulating peatlands to focus on WTD. Also, the use of general ecosystem models on peatlands often has an overarching goal of future incorporation into global climate models (e.g., Sulman et al., 2012), which deviates from the goals of field-based peatland restoration. However, organising a process-based ecosystem modelling system to analyse C fluxes in restored/rehabilitated peatlands is of interest especially during the period of time required to reach some form of ecosystem stability. If specific metrics are being developed to characterise restoring sites, such as estimating short-term methane emissions resulting from rewetting or longer-term vegetation community extents to incorporate in land-use change projections, it may be advantageous to maintain a "base" model with a hydrological site/region focus and subsequently vary a connected ecological model element relative to a desired output (i.e., "Model Combinations" like in Bernard-Jannin et al., 2018; Booth et al., 2022; Wilson et al., 2022). Compared to using a fully coupled model like *ecosys* or CoupModel, this approach may not require as much data, parameter calibration, time, and power to compute results.

## 5. Conclusions

The potential application of ecohydrological process-based models is a promising current and future field for exploring strategies for peatland restoration and evaluating post-intervention development over time. Modelling can contribute to the combined responsibilities of restoring peatland sites and tracking subsequent environmental impacts; this review highlights a number of models with existing applications to peatland contexts and potential for application to peatland restoration, though it is not exhaustive.

While this review has stressed the GHG-emissions sphere of modelling peatland environments ecohydrologically, vegetation community and water/soil nutrient outputs from ecohydrological models may be of additional value: it is probable that more process-rich models such as *ecosys* or CoupModel have the flexibility to produce outputs with implications for biodiversity or water quality as well, given that foundational biochemical processes are present. Water quality did not appear as a prominent area of research, where if modelling for a particular "end" had an environmental focus, it favoured GHG emissions (partly

caused by the specified search string, where GHG modelling was sought out specifically); additional review into water quality modelling as it relates to ecohydrology would be valuable. Predicting long-term peat accumulation (such as with DigiBog)

may be less compatible with models producing other desired outputs. It may still be valuable to be develop peat depth projections as a result of current restoration efforts, though this may need to remain a separate and supplementary focus to the more prominent ecohydrological component. Additionally, the introduction of vegetation on a previously bare landscape is (understandably) not included in most models reviewed and may need to be forced to mimic a restoration "event" over a period of time. Expansion upon existing models is active and ongoing in the field, for example with increasing the number of

dimensions in MPeat2D or adding new processes in MWMmic_NP. Finally, while the majority of this review has discussed process-based models, statistical (especially regression) models and ML provide additional ways to conceptualise and predict peatland processes; as requirements for spatial connectivity prove useful for policymaking or engineering design, newer research appears to diverge from site-by-site focus while still remaining distinct from scale-up global models.

Where current restoration efforts focus primarily on raising the WTD as a proxy for "successful" rewetting, here it may be

argued that evaluating the "success" of peatland restoration, especially for former industrial sites, must include the monitoring and projections for emissions, nutrient loading, and other ecosystem changes without falling into the trap of assuming the scientific consensus that improved hydrology tends to improve environmental impact. As such, there may be additional management options required for peatland restoration in the future which could alter the demands of ecohydrological modelling and go beyond the considerations from this review.

**Code/Data Availability**

The authors confirm that the data supporting the findings of this study are available within the article and its supplementary materials.

**Author Contribution**

MPS conducted the literature searches and all subsequent analysis. LG provided advice throughout the review process. MGH

proofread manuscript drafts to prepare for submission.

**Acknowledgements**

Thanks to Siya Shao for contributions with regard to MWM advancements. Many thanks to the reviewers for their valuable comments and criticisms.

**Competing interests**

The authors declare that they have no conflict of interest.

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

**Figures**

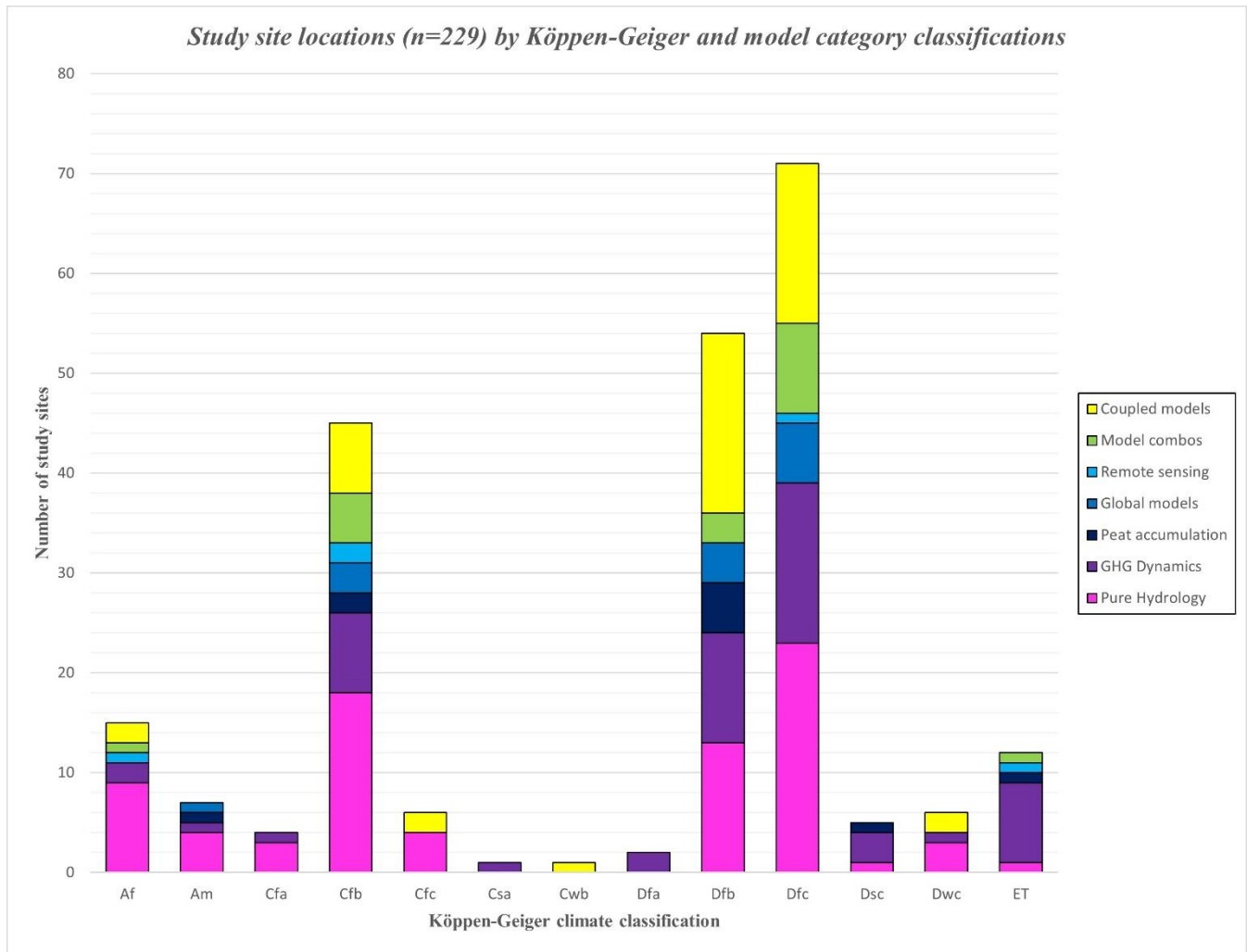

**Figure 1: Study site locations by Köppen-Geiger climate region. Much of NW Europe, a region of interest for this review's authors, is classified entirely as Cfb. For a full breakdown of climate region classifications and their abbreviations, see Kottek et al. (2006).**

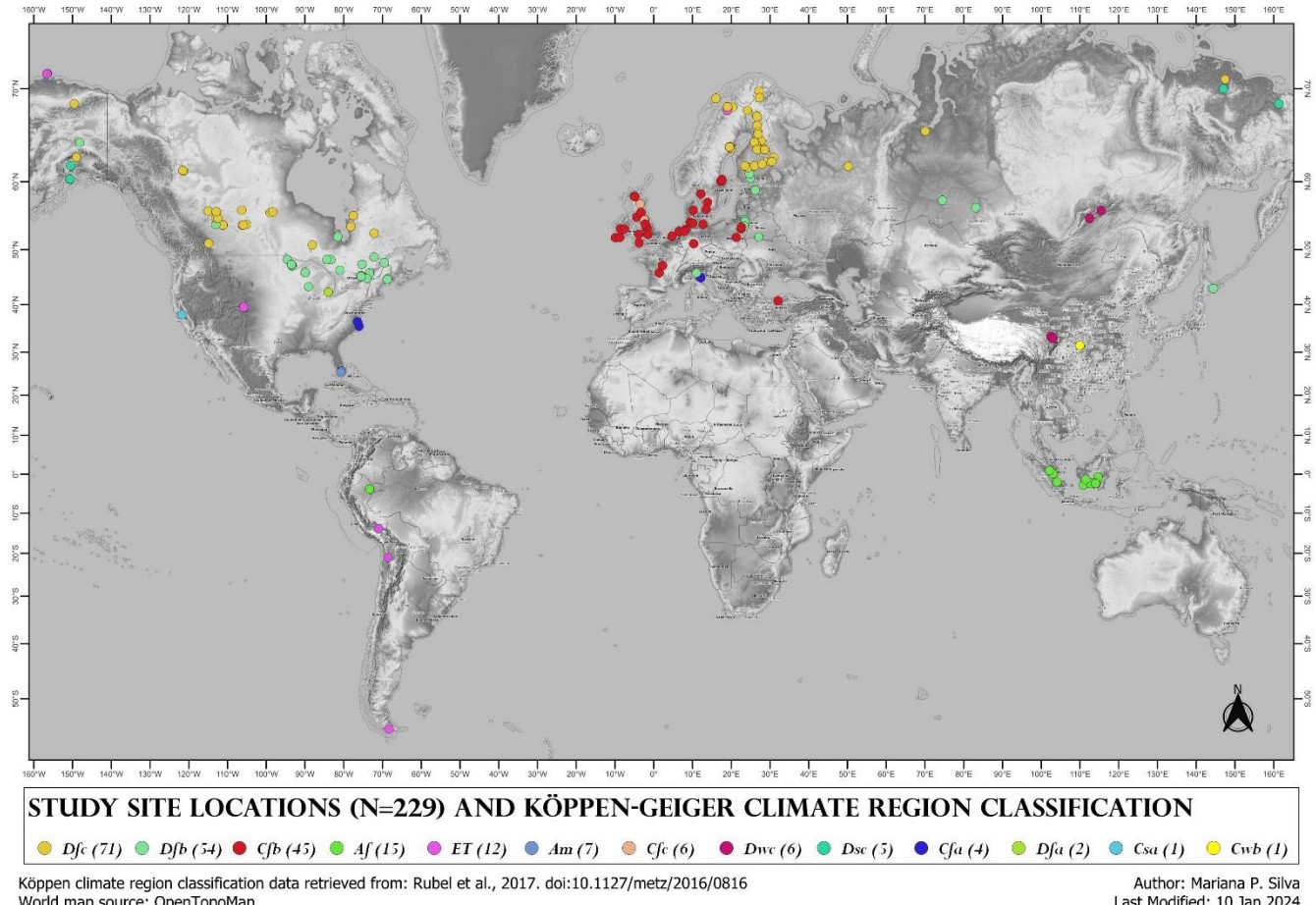

**Figure 2: Individual site locations for the 229 sites identified in papers from this review, with their corresponding Köppen-Geiger climate region classifications. Map generated using QGIS.**

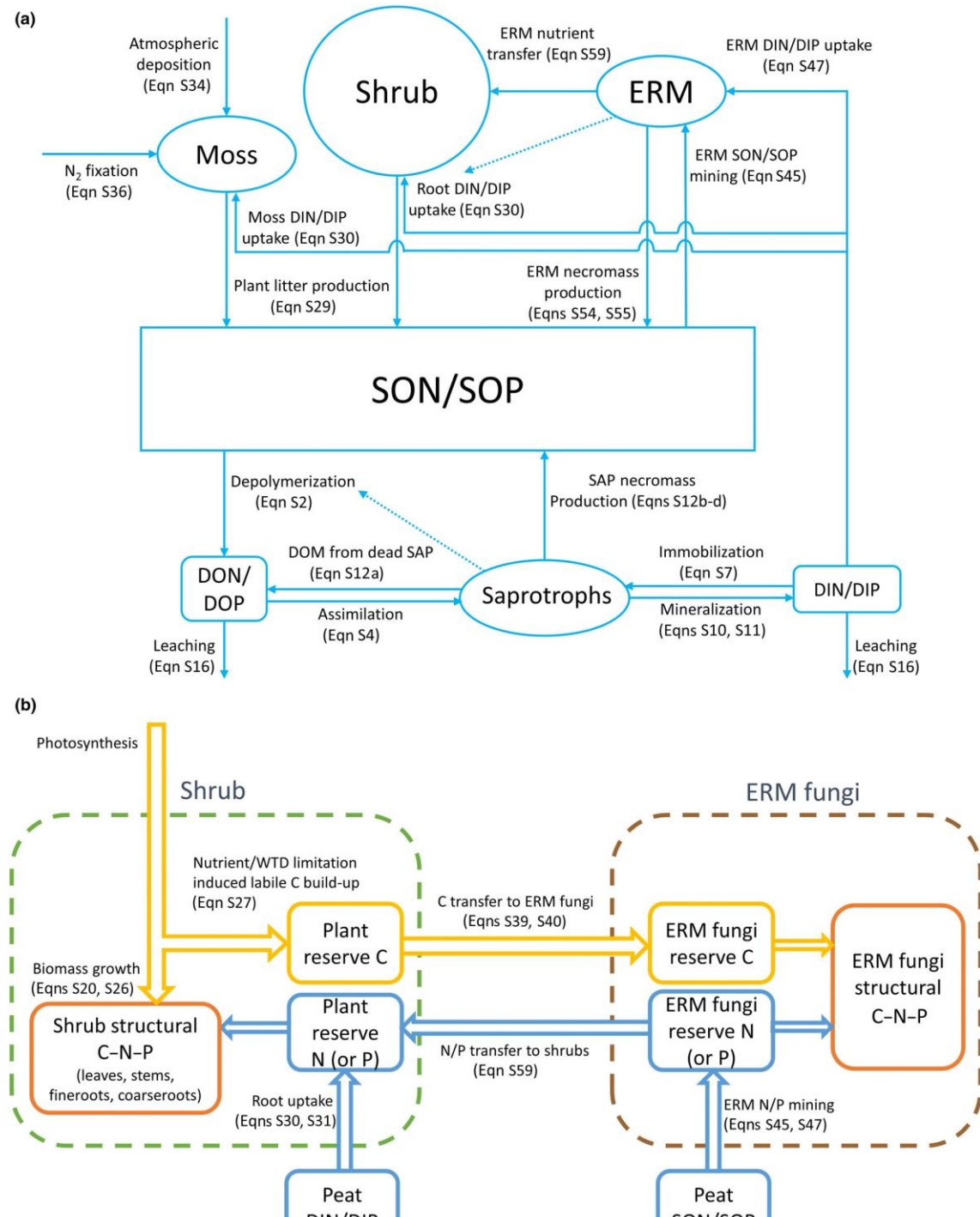

**Figure 3: Modelling framework for MWMmic_NP, showing (a) a schematic representation of the model structure and (b) a schematic of the Shrub-ERM carbon-nutrient exchange specifically. Details about abbreviations and notation can be found in Shao et al., 2023b.**

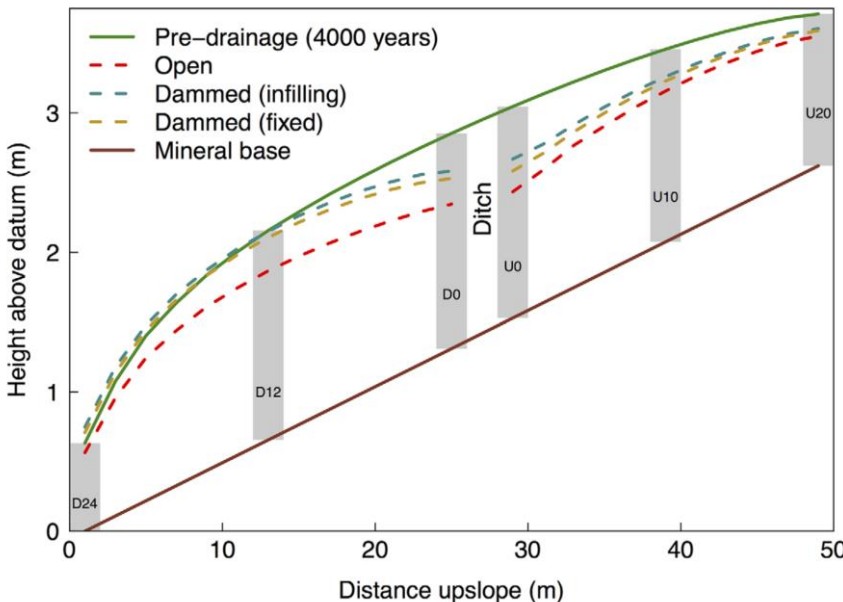

**Figure 4: Predicted peatland surface height before and after (300 years) ditch drainage and damming imposed upon a blanket bog transect, modelled using DigiBog by Young et al. (2017). Peat accumulation is modelled here alongside hydrology for the annotated peat columns (e.g., D24, D12); columns are identified using their position upslope (U) or downslope (D) of the ditch and the distance (m) of their edges that are nearest the ditch (e.g., column U10 occurs 10 m upslope of the ditch).**

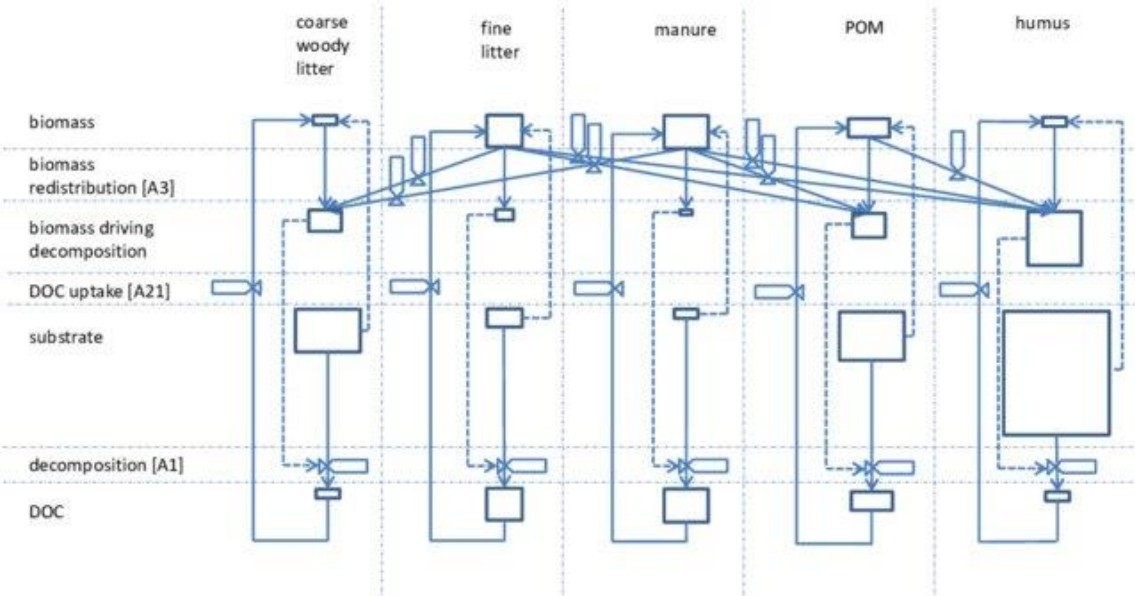

**Figure 5: A conceptual model of priming decomposition for N-rich humus in soils using *ecosys*. The transformations here are only a fraction of the full modelling framework for *ecosys,* but demonstrate the state variables and transfers within states which are used for biomass-substrate exchange in the model. Boxes represent state variables, sizes of which indicate relative sizes of biomass or substrate in each complex. Solid lines with valves represent rates of transfer among states; dashed lines represent drivers of these transfers. More details can be found in Grant, 2013.**

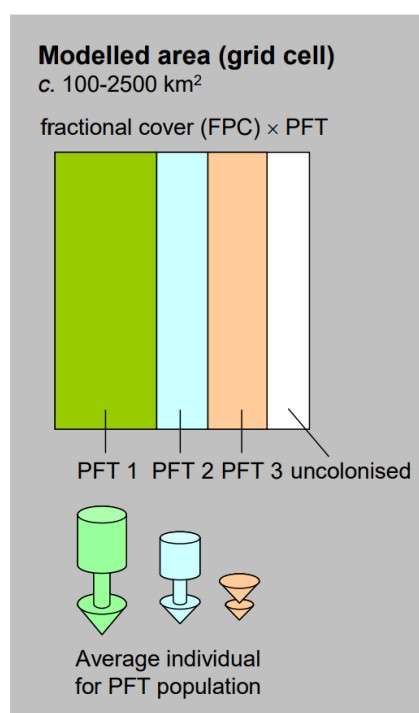
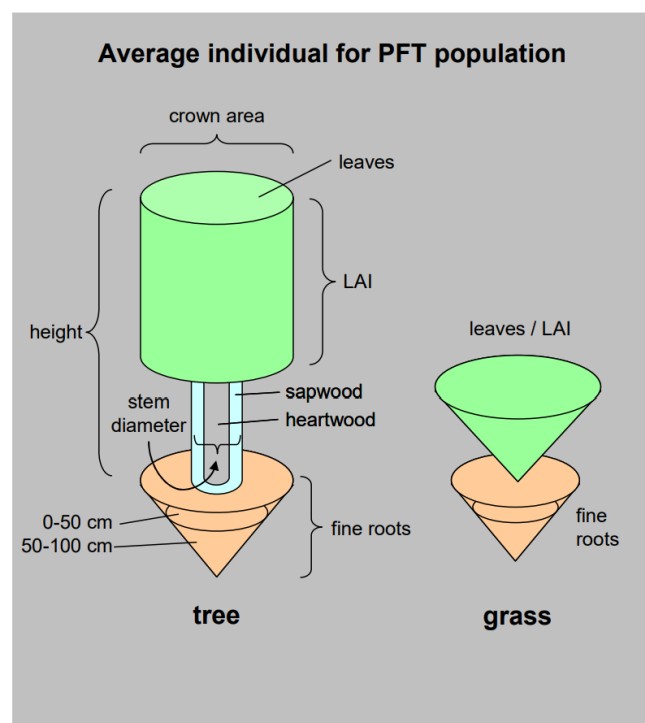

**Figure 6: The representation of vegetation as individuals and populations in LPJ-GUESS according to Plant Functional Types (PFTs). A "stand" option also exists to group canopy and groundcover vegetation.**

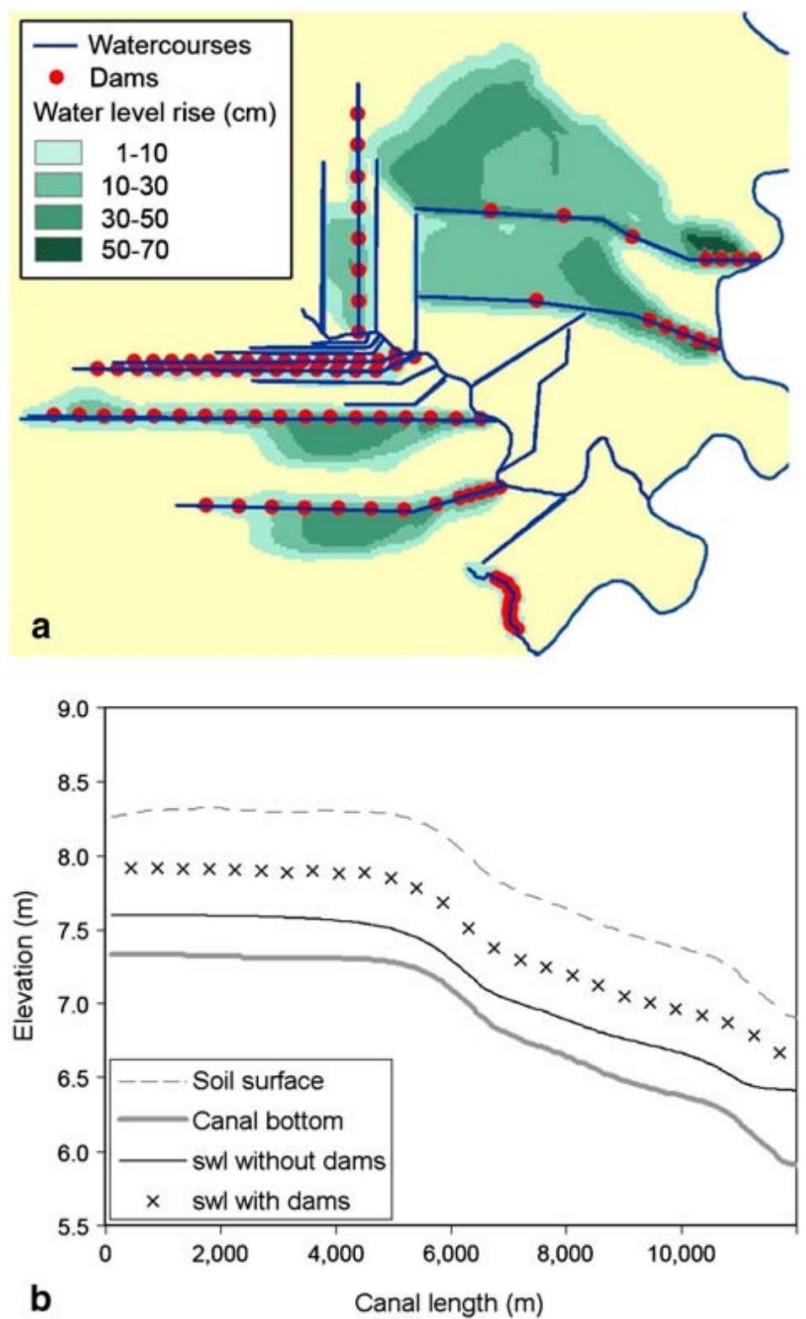


Figure 7: GIS-based predictions of water level rise for a tropical peatland using SIMGRO (Jaenicke et al., 2010). (a) Plan view of groundwater level rise after dam construction; (b) Surface water level rise (relative to soil surface elevation) in a single canal after dam construction.

**Tables**

 **Table 1: Details summarising common topics in papers removed from the dataset prior to analysis.**

| | Justification for removal | |
|---|---|---|
| | **Diverged from an ability to link with hydrology** | **Too specific** |
| 1st Pass | Ecological surveys of testate amoebae as proxies of palaeoenvironmental reconstructions | Permafrost dynamics |
| | Papers solely focussed on palaeoenvironmental reconstructions | Wildfire dynamics in forested peatlands |
| 2nd Pass | Vegetation dynamics with no link to water table depth | Digital elevation models used solely for mapping |
| | Peat hydraulics modelling carried out with lab samples rather than a full ecosystem | Tropical peat models with a distinct northern-latitude software counterpart |
| 3rd Pass | | Single-crop agricultural models (e.g., CERES for rice and DNDC for oil palm) |

**Table 2: Relative frequencies of named and unnamed models in this review.**

| Model category | # Named | # Unnamed | % Unnamed |
|---|---|---|---|
| Pure hydrology | 64 | 13 | 16.9% |
| GHG dynamics | 30 | 11 | 26.8% |
| Peat accumulation | 8 | 2 | 20.0% |
| Global models | 14 | 2 | 12.5% |
| Model combos* | 22 | 15 | 40.5% |
| Coupled models | 42 | 2 | 4.5% |
| Remote sensing** | 1 | 4 | 80.0% |
| **Total** | 181 | 49 | 21.3% |

*11 of 13 papers in this category list 2 models per paper. A total of 23 distinct models are counted here based on the information available from the papers.*

**Remote sensing papers featured prominently from a second post-hoc literature search; most are categorised in other categories if possible. However, 5 papers remained distinct in their objectives and were given their own category.*

**Table 3: Names of models recorded in database, condensed to show only those appearing more than once within a category, or once in a category where it exists more than once in another. Bolded models are discussed further in this review's discussion.**

| Category | Unique database name | Frequency |
|---|---|---|
| Pure Hydrology | **MODFLOW** | **15** |
| | Numerical model (unnamed) | 9 |
| | HydroGeoSphere (HGS) | 7 |
| | FEMMA | 5 |
| | TOPMODEL | 4 |
| | SWIFT2D | 3 |
| | **DigiBog** | **3** |
| | MIKE-SHE | 3 |
| | GIS-SAGA with LiDAR | 3 |
| | SIMGRO with GIS | 3 |
| | GEOTop (TOPMODEL-based) | 3 |
| | Mathematical model | 2 |
| | *ecosys* | 2 |
| | FLUSH | 2 |
| | SWAP | 2 |
| | HYDRUS (1 or 2D) | 2 |
| GHG Dynamics | *ecosys* | **3** |
| | DeNitrification DeComposition (DNDC) | 2 |
| | Numerical model (unnamed) | 2 |
| | PEATLANDVU | 2 |
| | **McGill Wetland Model (MWM)** | **2** |
| | **LPJ-GUESS/LPJ-WHyMe** | **2** |
| | ORCHIDEE-PCH4 | 1 |
| Peat Accumulation | Holocene Peat Model (HPM) | 2 |
| | **DigiBog** | **1** |
| Global Models | **PEAT-CLSM** | **3** |
| | MAgPIE | 2 |
| | ORCHIDEE-PEAT/ORCHIDEE-MICT | 2 |
| | LPX-Bern 1.0 | 1 |
| Model Combinations | Hummock-hollow (HH) model | 2 |
| | PERSiST and INCA-C | 2 |
| | NEST/NEST-DNDC | 2 |
| | FLUSH with a 1D sediment transport model (unnamed) | 2 |
| Coupled Models | *ecosys* | **6** |
| | **DigiBog** | **4** |
| | **CoupModel** | **5** |
| | RCG-C | 2 |
| | MILLENIA | 2 |
| | NICE-BCG | 2 |
| | **LPJ-GUESS** | **2** |
| | Wetland-DNDC | 2 |
| | **CLASS3W-MWM** | **2** |
| Remote Sensing | Unnamed | 3 |




**Table 4: Instances of use for most frequently used models, organised by Köppen-Geiger climate region classification. "Instances" may occur more than once within the same paper for different sites; here they are recorded as separate counts.**

| Köppen Classification | Frequently used models | | | | | | | Total per climate region |
|---|---|---|---|---|---|---|---|---|
| | CoupModel | Digibog | *ecosys* | LPJ | MODFLOW | MWM | PEAT_CLSM | |
| *no location listed* | | 4 | | 1 | 1 | | 2 | **8** |
| *Af* | | 1 | 2 | | 1 | | | 4 |
| *Am* | | | | | | | | 0 |
| *Cfa* | | | | | | | | 0 |
| *Cfb* | 2 | 4 | | | 2 | | 1 | **9** |
| *Cfc* | | | | | | | | 0 |
| *Dwc* | | | 2 | 1 | 1 | | | 4 |
| *Dfa* | | | | 1 | | | | 1 |
| *Dfb* | 2 | | 5 | 4 | 3 | 2 | 3 | **19** |
| *Dfc* | 1 | | 2 | 7 | 5 | 4 | 2 | **21** |
| *Dsc* | | | | | 1 | | | 1 |
| *ET* | | | 3 | 2 | | | | 5 |
| **Total per model** | 5 | *9* | *14* | *16* | **14** | 6 | 8 | |