# Peer review of "Reviews and Syntheses:* Evaluating the Potential Application of Ecohydrological Models for Northern Peatland Restoration: A Scoping Review"

_Biogeosciences, 2023_

## Author Response (AR4)

**Iteration 1**

**Reviewer 1**

Considering the long timescales required for peatland restoration and rehabilitation, modeling is critical for planning restoration and analyzing potential environmental impacts. This topic of this review is highly relevant for Biogeosciences. This review provides a comprehensive assessment of the current status of application of ecohydrological models on restoration of bogs and fens in northern regions considering a wide variety of potential environmental impacts. The authors clearly state their approach to identifying suitable studies and synthesize the results of these studies in a very useful manner. The results are based on 234 unique study sites which represents a substantial dataset. The authors identify the most widely used models, including LPJ, ecosys, and DigiBog. Much of the emphasis of the modeling exercises is on GHG emissions. The review highlights the emergence of remote sensing and machine learning to assess the progress of restoration with minimal intervention at sites. The figures and summaries provided in the review are very informative. This is an important review to provide context for peatland restoration.

This review is very timely as there are increasing efforts to restore and rehabilitate peatlands. The review synthesizes the models that have been applied to assess peatland restoration using several different categories. The conclusions are significant and important to understand the current and evolving status of this field of process-based modeling applied to peatlands in northern regions. The interpretations and conclusions are supported by the results of the review. The authors include detailed descriptions of some representative studies. The title is informative and reflects what is presented in the review. The abstract is excellent and summarizes the study clearly. The paper is clearly written. The review is concise and should not be shortened.

**[Changes made: emphasis on GHG emissions acknowledged especially in methods section when explaining the search string used. Additional mentions of the evident lack of water quality modelling research on peatlands also included, to reflect the statement made in lines 53-62.]**

**Reviewer 2**

The paper by Silva et al. provides a timely and much needed review of global modelling efforts as applied to peatland ecohydrology in a restoration context. This has the potential to be an excellent resource for future peatland modellers. While the abstract, introduction, methodology, and results are well-written, concise, and valuable, the discussion section offers little of value and is unfocused. The discussion section is rife with speculative statements, which as far as I can tell are unsupported assertions by the authors on the "likelihood" of future model developments and work that will be conducted. I personally do not feel that the discussion adds substance to the article, and should be reorganized and rewritten.

**[Changes made: As of now I have retained the current presentation and content of the discussion section. See below.]**

Two of the six figures are related to the spatial relationship between published peatland restoration studies and the Köppen-Geiger climate regions, while this has the potential to be an interesting feature of this review, it is not mentioned at all in the discussion section. The authors do not attempt to explain the patterns that the see, which perhaps are a spatial reflection of peatland prevalence, historical degradation, and being located within countries with a legacy of environmental stewardship and scientific funding. However, it is unclear what meaningful relationships can be derived from this analysis when the authors state that they deliberately placed an emphasis on NW Europe, as it is of particular interest to them. The remaining four figures are not well-integrated into the manuscript and do not contribute anything of

substance. I would recommend that they be removed and perhaps replaced with a compressed version of Table S1.

**[Changes made: I have added more detail to Figure 1 to improve the connection between location-based data and models included in the review. Some additional reflection on location-based data has been added in the discussion to add balance. Figure 2 is currently retained but could still be moved to supplementary material if deemed necessary. So far, figures 3-7 are retained in the manuscript to "break up" the text and provide visual context for models highlighted in the discussion. If it would be preferred by the editors, this can still be removed.]**

Arguably, the key themes that the authors identified as "emerging" reflect those that have already been established and firmly rooted in the contemporary scientific zeitgeist, and do not reflect truly emerging trends. The exception to this is the discussion on machine learning techniques. This does not represent a critical flaw but rather a missed opportunity to talk about where peatland science and modelling will go over the next decade.

**[Changes made: the term "emerg*" was searched for in the manuscript and wording was changed to more accurately represent established trends without speculating on new trends, excepting machine learning techniques.]**

My suggestion for the discussion section would be to categorize models by level of process complexity, and not restrict the discussion to (largely) ecosys, CoupModel, and DigiBog. The models HYDRUS, Hydrogeosphere, and MODFLOW (SURFACT) incorporate the equations governing unsaturated flow and transport. While this has the potential to reveal detailed process-based insights into peatland function and recovery, they tend to have prohibitive data requirements for most projects. In contrast, DigiBog has strengths that these complex finite-difference/element models lack being able to simulate peatland development on a far longer time scale (centuries-millennia). By organizing the discussion section along the spectrum of process complexity I think this review will be more valuable and integrate a larger number of studies and approaches, each with their own strengths and weaknesses. While I feel that this paper aligns with the aims and scope of Biogeosciences, the discussion section should be improved before the manuscript is accepted.

**[Changes made: this suggestion was not heeded as I believe that an overhaul of the discussion's structure is not needed. Instead, I was able to retain the structure of much of the discussion/results by removing the more speculative 'high potential' wording and instead highlighting the fact that these models highlighted did actually exist most frequently in the database within/across thematic categories (see line 250 and new Table 3). This means that while the review still does not enter into process-specific detail for a large number of studies, and process complexity is instead mentioned throughout the discussion within different thematic categories, there is increased justification for "restricting" the discussion to the few models highlighted.]**

Specific comments:

L33: What is a "governmental scale"? That covers everything from an individual municipality to a national effort.

**[Change made: the word "scales" was replaced with "spheres" to help clarify my point here.]**

L78: Two decades on from Belyea and Baird, the representation of feedbacks across spatiotemporal scales still proves a challenge – although some progress has been made, see Waddington et al. (2015)

Waddington, J. M., Morris, P. J., Kettridge, N., Granath, G., Thompson, D. K., & Moore, P. A. (2015). Hydrological feedbacks in northern peatlands. Ecohydrology 8(1), 113-127.

**[Change made: an additional note about progress was added to ensure accurate context, citing the reviewer's suggestion.]**

L232 The rationale for claiming some of these models have more applicability to northern peatland restoration is not clear to me. Furthermore, I'm not sure that I agree that some models have more or less relevance in this regard, they are simply used to understand different things.

**[Change made: statement changed to focusing on specific capabilities that may .]**

L244 "most natural" as in the largest number of processes?

**[Change made: clarification made replacing "most natural" with a phrase about the largest number of processes governing living organisms.]**

L245 According to S1, Putra et al. (2022) is a 2D model developed in DigiBog. Although I can see in Figure 7 that it is in fact 3D, however the model presented in Putra et al. (2022) is nearly axisymmetric and is not the most illustrative example of the value and capabilities of 3D modelling. Sutton and Price (2022), and Zi et al. (2016) are 3D.

**[Changes made: error rectified. An additional clarification including the reviewer's suggested citations was included.]**

L256 Although excellent models, the Melaku et al. (2022) paper seems to have only tangential relevance to restoration, similar to Hwang et al. (2018).

**[Change made: the example is retained, and I have included a disclaimer that this model's current applicability is only in the theoretical stage.]**

L261 Is direct coding required for anything other than process modification? Please expand on the "etc."

**[Change made: sentence was restructured, and additional context/reflection added (now beginning on line 288).]**

L268 This is incorrect and not the definition of the acrotelm. Besides being a potentially outdated conceptual framework (see Morris et al., 2011), the acrotelm comprises more than the "living layer" in bogs and includes partially and moderately decomposed plant matter (although plant roots can traverse the boundary between acrotelm and catotelm - as there is often more than just Sphagnum mosses living in peatlands). Furthermore, while the catotelm may comprise more well-decomposed peat with typically lower hydraulic conductivity, it is not hydrologically inactive and performs crucial hydrological functions in peatlands.

Morris, P. J., Waddington, J. M., Benscoter, B. W., & Turetsky, M. R. (2011). Conceptual frameworks in peatland ecohydrology: looking beyond the two-layered (acrotelm–catotelm) model. Ecohydrology 4(1), 1-11.

**[Changes made: I referenced the term as a way to show where models are progressing from**

(retaining the words "changes occurring in" and adding "past designations of …"), citing Clymo in 1978 and Morris et al. in 2011 for comparison.]

L272-274 This is just my opinion, but I think more restraint should be exercised when recommending the use of models to practitioners that may not be aware of the myriad assumptions, limitations, and caveats that can apply to a model. That has the potential to cause more harm than good.

[Change made: Rephrased the sentence to focus more on the ease of execution (with fewer inputs and access to source code for greater context) rather than claiming no background knowledge is needed, and removed identification of potential target audiences.]

L277 This may not be germane to the overall paper, but it should be noted that fen-bog transitions can also be instigated quickly due to sudden drops in water table.

[Changes made: added a discussion of this point in lines 305-309.]

L277-279 These two sentences are rather hypothetical/speculative and unnecessary.

[Changes made: combined with discussion mentioned above to become less speculative.]

L297-298 One of the consequences of the organization of the discussion section (and focus on the Digibog, CoupModel, and ecosys) is that important developments are portrayed as unanswered open questions. The work of McCarter and Price have investigated this in a peatland restoration context using models.

[Changes made: cited Gauthier, McCarter, and Price (2018) as well as Lehan et al. (2022) to provide additional context for peat hydraulic behaviour during restoration.]

L390 This is written as though this is unknowable information, but surely this is made clear within the paper, if not perhaps it is not the best example to use.

[Changes made: shortened information to only briefly acknowledge the oil sands mining context.]

L394 The concept of an HRU does not inherently limit the resolution that different wetlands could be represented, many semi-distributed hydrologic models (ex. Raven) can have an arbitrarily large number of HRU, such that they begin to more closely resemble a fully spatially-distributed model.

[Change made: rephrased "HRUs" to "the number of currently incorporated HRUs".]

L414-415 What makes it likely that a 13 year old paper will have further modelling occur if it has not already? Why speculate in this manner? There may be better examples to draw upon to describe the use of 3D modelling in a peatland restoration context.

[Response: included to highlight that it is still rare that simulations combine 3-D modelling with a specific restoration context like in Jaenicke et al. (2010), while disclaiming that the study is old.]

Technical corrections:

L142: Missing the word "is"

**[Rectified.]**

**Iteration 2**

**Reviewer 1**
[no new changes]

**Reviewer 2**

I appreciate the changes made by the authors to address my specific comments. However, I still reiterate my initial stance that the discussion section does not represent an insightful and meaningful contribution to the paper. I maintain that the manuscript would be greatly improved by a reorientation of the discussion section to process-based complexity in the relevant modelling platforms.

**(For context, below is what was suggested in the first round)**

> "…The discussion section is rife with speculative statements, which as far as I can tell are unsupported assertions by the authors on the "likelihood" of future model developments and work that will be conducted. I personally do not feel that the discussion adds substance to the article, and should be reorganized and rewritten…
>
> …My suggestion for the discussion section would be to categorize models by level of process complexity, and not restrict the discussion to (largely) *ecosys*, CoupModel, and DigiBog … By organizing the discussion section along the spectrum of process complexity I think this review will be more valuable and integrate a larger number of studies and approaches, each with their own strengths and weaknesses."

**[Response: I concede that while we increased our effort to demonstrate the relevance of themes corresponding to different modelling ends, organising the discussion around the means of modelling by focussing on process-complexity will provide added benefit to readers.]**

**[Changes made: the discussion was reworked extensively, where much of the thematic analysis is retained in a "synthesis" portion at the end. Tables and supplements related to "top three" models were removed, and new figures outlining examples of model schematics are included instead of some model outputs to provide additional context.]**

Specific Comments:

- L237-245 duplicated lines

- L312-316 duplicated lines

**[Response: On my end, I do not see duplicated lines. Perhaps it was a downloading glitch. If this issue persists, please let me know.]**

**Thanks again for these criticisms and I hope the overall result is a satisfactory improvement to the review. --M.P.S.**

**Iteration 3**

*Editor's comments.*

The review of Silva makes a number of interesting points but I feel that a few key changes need to be made to avoid overly critiquing other work and questioning the utility of key figures at the expense of a more concise summary of the extensive literature review that was undertaken.

The comment on line 70: 'Yet, despite decades of research, models of this kind are deficient in addressing the entirety of restoring peatlands (i.e., the degradation, mid-restoration, and long-term impact stages) in an efficient, ecohydrological manner.' is to me a bit subjective and overly-critical of the models which may not be explicitly designed for restoration.

- **The term "deficient" is indeed a bit subjective. I was hoping to explain concisely that the *completeness* with which bog restoration is modelled is lacking, because individual modelling projects provide disjointed simulations of either past ecology, current hydrology, future global impacts, or other phenomena—but this information when synthesised cannot easily represent the whole restoration process where *eco*hydrology is the goal, especially when longitudinal studies are also less common.**
- **I have reworded the statement to reflect this while sounding less critical:**
  - **'Yet, despite decades of research, models of this kind *rarely* address the entirety of restoring peatlands (i.e., the degradation, mid-restoration, and long-term impact stages) in a *complete*, ecohydrological manner.'**

Section 1.1 and elsewhere: when possible, try not to make the authors the subject of a sentence or else the narrative begins to sound disjointed with the topics being many different people instead of how these people have created new knowledge. This admittedly can be a bit hard to do in a review paper, but a bit of work toward this end would improve the flow.

- **This is understandable. While Section 1.1 reads a but more like a narrative than other sections, I will attempt to avoid this format for sentences across the board.**
- **This comes up a bit later in the discussion as well when describing peatland-related work carried out by different models. Most lines in this section were also adjusted.**

105: 'neglects' is a bit too critical if restoration wasn't the focus.

- **This is valid – here I was again going for conciseness and choosing one slightly more critical word compared to a number of smaller words.**
- **I have changed to wording to "does not include".**

Table 4 wasn't referred to in the text and could go a long way to help synthesize results, which struck me as a bit short and missed an opportunity to provide a more synthetic analysis. I

understand that the conventional categories of 'Results' and 'Discussion' can be difficult to fit in to the flow of a review manuscript, but given the literature review I feel that they could be.

- **This was a mistake from the previous revision; I accidentally took a chunk out referencing Table 4 that should have been retained.**
- **I have inserted this section and provided some additional remarks.**

Are Figures 2-7 from other papers and if they have already been published is there permission to re-publish? I don't feel that any of these are necessary to describe results although an exception could perhaps be made for Figure 4 and perhaps Figure 7 but the latter seems like it may be a re-print. Looking at Figure 10 in Jaenicke et al. it is, but the work is available under a CC noncommercial license so it can be used. I guess I'm just confused as to why many of the figures are necessary and some like Fig. 5 aren't rendering on the page in sufficient quality although this may be due to the word processing program rather than the native resolution of the figure.

- **I was considering this myself – but I wasn't sure if it may be visually helpful for readers to break up the discussion of myriad model details with diagrams showing how some of them may operate.**
- **In terms of permissions, as far as I could tell for all of these diagrams it appeared that citing the authors when describing the images would be sufficient, and for the LPJ diagrams I recall the model's website stipulated that no copyright or citation was necessary to use the images.**
- **After re-reading the final discussion, I believe Figures 4 and 7 could be retained to provide illustration of restoration-specific modelling, which is what I've done for this resubmission.**
- **However, if it is too much of a hassle to chase down permissions, or if it may appear confusing to readers, I am happy to remove all figures after Figure 2.**

**Thanks very much for your suggestions and criticisms to bring this paper to a stronger and more informative level. --MPS**

**Final iteration: file validation**

*Editorial support team comments:*

For now, we will proceed with your manuscript as submitted. However, please adjust your manuscript files before your next file upload (next round of revision or after acceptance) considering the following requirements:

1. With the next revision, please re-check content of the supplement. It seems, that it not contain additional materials, but figures and tables from the manuscript. Please note that you will be asked to provide these materials at other stages of the review process.

**[Rectified.]**

2. Regarding figure 4: please ensure that the colour schemes used in your maps and charts allow readers with colour vision deficiencies to correctly interpret your findings. Please check your figures using the Coblis – Color Blindness Simulator ([https://www.color-blindness.com/coblis-color-blindness-simulator/](https://www.color-blindness.com/coblis-color-blindness-simulator/)) and revise the colour schemes accordingly.

**[After checking Figure 4 as suggested through the colour blindness simulator, I believe there is enough distinguishability between symbols and colour gradation that the figure can go unchanged. I also believe the figure should remain unaltered because it is the result from a previous study (cited in the caption) and it seems simplest to preserve it in the way it was originally published.**

**The figure will go unchanged for the final upload but I will be sure to note this response in my Author's Response file, in case further changes are still deemed necessary in validation.**

**Thank you – MPS]**